# The relationship between modifiable lifestyle behaviours and self-reported health in children and adolescents in the United Kingdom

**Jason Moran** \*, **Gavin Sandercock, Brandon S. Shaw** , **Paul Freeman, Catherine Kerr, Ina Shaw**

School of Sport, Rehabilitation, and Exercise Sciences, University of Essex, Colchester, United Kingdom

\* jmorana@essex.ac.uk

**Data Availability Statement:** Data were obtained from the UK Data Service at the University of Essex. The study sample consisted of 11,859 (5941 boys, 5918 girls) adolescents who took part

## Abstract

Obesity, along with its associated health issues, is closely tied to lifestyle habits. While certain elements affecting childhood health, such as genetics and ethnicity, are beyond individuals' control, there exists modifiable lifestyle behaviours that can facilitate healthier living. This study employed multiple regression analysis to investigate the relationship between specific modifiable lifestyle behaviours and self-reported health. The independent variables considered included days of moderate to vigorous physical activity (MVPA), fruit and vegetable consumption, breakfast frequency, school night sleep duration, and non-school night sleep duration. These variables were chosen for their practical modifiability within participants' daily lives. The analysis revealed a highly significant overall model ($F(13,11363) = 191.117$, $p < .001$), explaining 17.9% of the variance in self-reported health. Notably, higher MVPA levels were associated with improved self-reported health ($B = 0.136$ to $0.730$, $p < .001$). Additionally, regular breakfast consumption and increased fruit and vegetable intake exhibited positive associations with self-reported health ($B = 0.113$ to $0.377$, $p < .001$), while girls reported lower self-reported health ($B = -0.079$, $p < .001$). School night sleep duration was positively linked to self-reported health ($B = 0.071$, $p < .001$). Furthermore, a dose-response relationship between MVPA, dietary habits, and health was identified. These findings hold substantial potential for public health campaigns to promote healthy behaviours and prevent chronic diseases in young individuals. It is imperative to emphasise that all the variables considered in this study are readily modifiable aspects of individuals' lives, offering a promising avenue for personal health and well-being enhancement.

## Introduction

The prevalence of obesity and its comorbidities have been associated with children's and adolescents' lifestyle habits, such as exercise and eating patterns, which are commonly determined by parental influence [1]. Failure to engage in such health-promoting habits can have negative

in the 2015 sweep of the United Kingdom's Millennium Cohort Study. The datasets generated during and/or analysed during the current study are available at https://beta.ukdataservice.ac.uk/datacatalogue/series/series?id=2000031.

**Funding:** The author(s) received no specific funding for this work.

**Competing interests:** The authors have declared that no competing interests exist.

effects on a number of key indicators of health status such as body mass and blood cholesterol [1]. Though there are numerous unmodifiable factors that can negatively affect health during youth, such as genetics or ethnic characteristics, there are also some modifiable lifestyle behaviours that can underpin a healthy lifestyle and longer lifespan [2]. For example, some of the modifiable factors that can influence a youth's health include their nutritional habits, and physical activity profile, as well as both the quality and quantity of sleep they attain [2]. Factors such as sex and genetic characteristics are unmodifiable, however, in most western countries at least, control of dietary and exercise factors, as well as sleeping habits, are mostly determined by a person's own free choice. In a similar vein, other important factors such as breastfeeding can positively affect obesity status [3] but despite being readily modifiable [2], personal agency plays almost no role in whether or not an infant is breastfed.

Given the above, there are a number of influential factors that can inform lifestyle-related decisions both with or without parental influence, or other extraneous, influence. This is important information for health professionals to know because tools such as promotional campaigns to enhance youths' health have not been shown to be overly successful with those that transgress on a young person's desire for independence most likely to fail in their objectives [4]. Similarly, convoluted messages about the relative benefits and harms associated with numerous foods and habits makes it difficult for consumers to make sound nutritional and lifestyle choices [5]. Accordingly, engagement in a small and manageable number of beneficial lifestyle habits could be advantageous in underpinning good health in youths. The importance of achieving this aim has become more urgent in recent times with claims that the simultaneous engagement in multiple online activities has reduced youth's ability to devote concentration to singular tasks, hampering their overall attention span [6]. This makes it more crucial than ever that messages pertaining to healthy living habits are reinforced in a brief, yet informative, manner to this population.

Hover, despite highly encouraging evidence, it remains unclear as to what the combined effect of modifiable lifestyle behaviours is when they are considered as part of a unified strategy to underpin health in young people. During adolescence, modifiable habits, such as engagement in exercise, can become highly erratic with lower levels of physical activity and greater autonomy over the types of food one eats becoming more prevalent at this time [7]. Moreover, factors such as poor dietary and sleep habits, as well as low levels of physical activity, can also negatively affect maturation of the adolescent brain, negatively impacting social, emotional, and cognitive development as a youth transitions to adulthood [8]. On this basis, the objective of this study was to therefore determine the interdependence of modifiable lifestyle behaviours, such as the regular eating of breakfast, fruit and vegetable intake, engagement in MVPA and sleep time on self-reported health in British adolescents.

## Methods

This sample consisted of 11,859 (5941 boys, 5918 girls) adolescents who took part in the 2015 sweep of the United Kingdom's Millennium Cohort Study (MCS) [9]. Participants in this study were first examined at the age of nine months, in 2001, and at six separate sweeps after that, up to 2018. In 2015, the children in this study were 14 years old, an age we chose due to the substantial changes in behaviour and brain development that are observed at this time [10]. Data were obtained from the UK Data Service at the University of Essex. The MCS received ethical approval from the Research Ethics Committee of the National Research Ethics Service. Participants and parents provided their consent to take part in the surveys. Both participants and parents provided their written consent to take part in the surveys [11]. The data

were accessed for research purposes on 30[th] March, 2023. None of the authors could identify individual participants during or after data collection.

The dependent variable in this study was self-reported health which was evaluated with categorical answers to the question "How would you describe your health generally?" (1 = Excellent; 2 = Very good; 3 = Good; 4 = Fair; 5 = Poor). The explanatory variables were breakfast eating frequency (Q: How often do you eat breakfast over a week?; A: 1 = Never; 2 = Some days, but not all days, 3 = Every day), daily fruit consumption (Q: How often do you eat at least 2 portions of fruit per day?; A: 1 = Never; 2 = Some days, but not all days, 3 = Every day),), daily vegetable intake (Q: How often do you eat at least 2 portions of vegetables including salad, fresh, frozen or tinned vegetables per day?; A: 1 = Never; 2 = Some days, but not all days, 3 = Every day), moderate to vigorous physical activity (MVPA) engagement (Q: On how many days in the last week did you do a total of at least an hour of moderate to vigorous physical activity?; A: 1 = Every day; 2 = 5–6 days; 3 = 3–4 days; 4 = 1–2 days; 5 = Not at all) and sleeping habits (total sleeping hours converted to a composite scale variable based on questioning of participants' sleep and wake times on school and non-school days). Predictors were selected *a priori* and were chosen based on the rationale that they were additive actions that an individual could actively take to improve their health.

## Statistical analysis

This regression analysis was performed in JASP version 0.11.1, University of Amsterdam. A linear multiple regression analysis was executed with self-reported health entered as the dependant variable. The independent variables were number of days of MVPA in the last week, fruit intake, vegetable intake, numbers of days on which breakfast is eaten, number of hours of sleep on school nights and number of hours of sleep on non-school nights. These variables were selected on the basis that they are factors that are modifiable within the participants' lives. To determine the fit of the regression model $R^2$, adjusted $R^2$ and the root mean square error (RMSE) were evaluated. Multicollinearity was examined with reference to the variance inflation factor (VIF) which must have been <4 for a variable to be retained in the model. Collinearity statistics, coefficients covariance matrix, and variance proportions were also examined to check for multicollinearity among the independent variables. Additionally, the Durbin-Watson statistic was used to test for autocorrelation in the residuals. The standardised beta coefficients were used to determine the effect sizes of the predictor variables in the regression analysis. These effect sizes were classified according to Cohen's criteria which categorise standardised coefficients of 0.10, 0.30, and 0.50 as being 'small', 'medium', and 'large' effects respectively [12]. The significance level for all statistical tests was set at $p < 0.05$.

## Results

Table 1 displays descriptive data for the participants.

To determine the validity of the predicted model, relevant plots and graphs were visually inspected for normality, independence, linearity, homoscedasticity, and multicollinearity. No violations of these assumptions were observed, which suggests that the predicted model was an appropriate fit for the data.

The results of the linear regression analysis which evaluated the effects of diet, MVPA, and sleep on a measure of self-reported health in adolescents are presented in Table 2. The model incorporates a number of predictors, such as MVPA, breakfast consumption, fruit and vegetable intake, sleep hours on school and non-school days and sex. The overall model was statistically significant ($F_{(13,11363)} = 191.117$, $p < .001$) and accounted for 17.9% of the variance in

**Table 1. Descriptive data for the participants.**

| Frequencies for study variables | | |
|---|---|---|
| **Age (yrs)** | **Frequency** | **Percent** |
| 13 | 2858 | 24.10 |
| 14 | 8841 | 74.55 |
| 15 | 160 | 1.35 |
| **Sex** | | |
| Boy | 5941 | 50.10 |
| Girl | 5918 | 49.90 |
| **Health (5 is highest)** | | |
| 1 | 221 | 1.93 |
| 2 | 1295 | 11.30 |
| 3 | 4242 | 37.02 |
| 4 | 4266 | 37.23 |
| 5 | 1436 | 12.53 |
| **MVPA (5 is highest)** | | |
| 1 | 506 | 4.40 |
| 2 | 2774 | 24.14 |
| 3 | 3872 | 33.70 |
| 4 | 2222 | 19.34 |
| 5 | 2117 | 18.42 |
| **Breakfast consumption (days)** | | |
| 1 | 987 | 8.60 |
| 2 | 4321 | 37.66 |
| 3 | 6166 | 53.74 |
| **Fruit intake (days)** | | |
| 1 | 994 | 8.67 |
| 2 | 6939 | 60.54 |
| 3 | 3528 | 30.78 |
| **Vegetables (days)** | | |
| 1 | 911 | 7.96 |
| 2 | 6255 | 54.65 |
| 3 | 4280 | 37.39 |

self-reported health, as indicated by the adjusted R-squared value of 0.178. The root mean square error (RMSE) of the model was 0.831.

Table 2 displays the coefficients for each predictor. All predictors were found to be statistically significant, with the exception of sleep hours on non-school days. Higher levels of MVPA were associated with better standardised coefficients for self-reported health (ES = 0.059 to 0.297, p < .001). Breakfast consumption and fruit and vegetable intake were also positively associated with self-reported health (ES = 0.047 to 0. 0.138, p < .001) while girls reported marginally lower levels of self-reported health (β = -0.079, p < .001). Furthermore, sleep hours on school days had a positive relationship with self-reported health (ES = 0.008, p < .001), though the standardised coefficient was very low.

Based on the presented beta coefficients, it can be seen that there was a significant dose-response relationship between self-reported health and the chosen predictors with higher levels of MVPA and consumption of breakfast, fruits, and vegetables being associated with better self-reported health. Specifically, the beta coefficients for MVPA were 0.136 for 1–2 days of exercise, 0.334 for 3–4 days, 0.520 for 5–6 days and 0.730 for exercising every day, indicating a

**Table 2. Regression coefficients and collinearity statistics for predictors of self-reported health among adolescents.**

| Model | Predictor | Unstandardized Coefficient | Standard Error | Standardized Coefficient | t-value | p-value | Tolerance | VIF |
|---|---|---|---|---|---|---|---|---|
| $H_0$ | (Intercept) | 3.472 | 0.009 | | 404.163 | < .001 | | |
| $H_1$ | (Intercept) | 1.897 | 0.096 | | 19.665 | < .001 | | |
| $H_1$ | MVPA (2) | 0.136 | 0.041 | 0.059 | 3.347 | < .001 | | |
| $H_1$ | MVPA (3) | 0.334 | 0.04 | 0.144 | 8.361 | < .001 | | |
| $H_1$ | MVPA (4) | 0.52 | 0.042 | 0.21 | 12.447 | < .001 | | |
| $H_1$ | MVPA (5) | 0.73 | 0.042 | 0.297 | 17.336 | < .001 | | |
| $H_1$ | Breakfast (2) | 0.116 | 0.03 | 0.048 | 3.883 | < .001 | | |
| $H_1$ | Breakfast (3) | 0.33 | 0.03 | 0.138 | 10.978 | < .001 | | |
| $H_1$ | Fruit (2) | 0.113 | 0.03 | 0.047 | 3.735 | < .001 | | |
| $H_1$ | Fruit (3) | 0.311 | 0.034 | 0.115 | 9.191 | < .001 | | |
| $H_1$ | Vegetables (2) | 0.173 | 0.031 | 0.073 | 5.531 | < .001 | | |
| $H_1$ | Vegetables (3) | 0.377 | 0.034 | 0.138 | 11.163 | < .001 | | |
| $H_1$ | Sex (Girl) | -0.079 | 0.016 | -0.105 | -4.861 | < .001 | | |
| $H_1$ | School day sleep hours | 0.071 | 0.008 | 0.081 | 8.936 | < .001 | 0.947 | 1.055 |
| $H_1$ | Non-school day sleep hours | 2.10E-04 | 0.007 | 2.83E-04 | 0.032 | 0.974 | 0.947 | 1.055 |

strong positive relationship between MVPA and self-reported health. In a similar vein, the beta coefficients for breakfast consumption, fruit consumption, and vegetable consumption increased with each level, thus indicating a positive relationship with self-reported health.

## Discussion

The objective of this study was to evaluate the multivariate relationship between self-reported health and modifiable lifestyle factors in British adolescents who partook in the Millennium Cohort Study, specifically the fifth sweep in 2015. To summarise, the analysis revealed that MVPA, fruit and vegetable intake, as well as breakfast consumption, are all important additive predictors of health, with MVPA being the most important amongst these. Though the model accounted for only 17.9% of the variance in self-reported health, almost all of the included predictor variables shared a significant positive relationship with the independent variable. The standardised beta coefficients from the regression analysis indicate that MVPA, breakfast and intake of fruit, and vegetables all had a positive and significant effect on self-reported health, with MVPA, conducted every day, showing the largest standardised effect (0.297 [small to medium]). Though no individual variable exerted a particularly large effect on self-reported health when considered in isolation, as evidenced by the R-squared value of 0.178, these factors may represent a meaningful combined strategy for health improvement. The results also seem to be suggestive of a potential dose-response relationship between these health modifiable behaviours and self-reported health in British adolescents.

The findings of this study could have important implications for promoting positive lifestyle habits and improving health outcomes in adolescents in the United Kingdom. They appear to indicate that readily-modifiable lifestyle factors, despite playing an individually small role in supporting young individuals' health, may exert an important synergistic influence on wellbeing. For example, children and parents can easily share in decisions on the amount of MVPA that is executed within a weekly timeframe. Similarly, the choice to concurrently consume optimal quantities of both fruits and vegetables, and include breakfast eating in a daily schedule, is within the capability of most individuals. Accordingly, this affords children and their parents alike, a degree of collective control over their health status.

That MVPA was found to have the largest standardised effect on self-reported health status is not entirely surprising. In children, MVPA has previously been associated with reducing weight gain [13], lowered cardiometabolic risk factors [14], reduced BMI [15] and greater mental wellbeing [16]. However, despite the relative ease in implementing some or all of the behaviours described in the current study, it is curious that childhood ill-health continues to rise in different regions of the world with patterns of obesity in high-income Anglophone countries now being replicated across east and south Asia [17]. Furthermore, recently, there has been an observed trend in reduced adolescent health-related fitness which has coincided with a decline in the provision of physical education in the United Kingdom [18]. This decline has been experienced across a wide cross-section of youths in the country, most worryingly in those with lower fitness levels, a subsection already vulnerable to the risks of a sedentary lifestyle. Whilst decisions over the provision of physical education are made at the level of the policy-maker, parents can work with their child to control the readily-modifiable habits that were used as predictor variables in this investigation. Previous work [19] has shown that parental support is a vital element in encouraging physical activity engagement in young people, either directly or indirectly, by enhancing one's self-confidence. Similarly, the notion of the "athlete family" identity has previously been outlined with such systems described as those which invest a considerable amount of time, money and emotional support towards youth sports endeavours [20]. In the current economic climate, the investment of money into these types of activities may not be realistic for all families, however, investment of time and support may represent more accessible actions for families to encourage uptake of sport and physical activity in adolescents. On this basis, initiatives to promote healthy behaviours in youths should focus on parents' ability to provide practical and motivational encouragement for their children to engage in more exercise and to adopt health eating and sleeping habits [19].

Research-evidenced initiatives of this nature, that families could find effective, could include meal planning, collective bouts of physical activity, food preparation and gardening activities. Meal planning involves collaboration between parents and children in choosing recipes, creating appropriate shopping lists and preparing meals together can enhance children's nutritional behaviours [21]. In addition to this, parents and children who prepare healthy snacks together can potentially increase their intake of fruits and vegetables [21]. Similarly, collaborating on growing their own fruits and vegetables in a garden has previously been shown to increase children's intake of fruits and vegetables and to further inform their attitudes towards healthy eating behaviours [22]. Moreover, engaging in physical activities together, such as going walking or cycling, has been found to underpin higher physical activity levels in children, reducing sedentary behaviour in the process [23]. All of these interventions together are generally costless to engage with and can be executed readily by parents and children within certain time constraints. These activities are also supported by research as being effective strategies for the improvement of dietary habits and increasing physical activity levels in young populations.

The results of this study also imply that individuals who consume breakfast on a more frequent basis may experience improved health as a result. This is consistent with previous work that showed that breakfast consumption was associated with a range of positive markers of health including improved cognition, enhanced weight management and a lowered risk of chronic disease [24]. As the nutritional behaviours that are established during the formative years of childhood and adolescence are likely to persist into adulthood, it is crucial to identify the factors that influence breakfast-eating habits in children and adolescents, thus reinforcing engagement in healthy behaviour across the lifespan. Accordingly, the family environment plays a significant role in determining the dietary activity of the young person however, despite this, breakfast appears to be the meal that is skipped more often than any other during the day

[25]. A systematic review [25] reported that parental breakfast-eating habits were consistently associated with adolescent habits and this is an important factor for health professionals to consider. In line with the recommendation of the current study, this suggests that parents should make efforts to act as the primary positive role model in the shaping of behaviours by adopting healthy breakfast habits such that they are repeated by their children.

Similar to above, the results of this investigation imply that fruit and vegetable consumption may be a important behaviour in promoting healthy outcomes in adolescents. This aligns with previous research that demonstes the positive effects of fruit and vegetable intake on a range of different health markers such as better cardiovascular health, recued risk of chronic diseases and enhanced mental health [26–28]. Even though we reported a dose response for the potential effect of fruit and vegetable consumption on self-reported health, both parents and health professional should be aware that there may be an upper limit of benefit in this, and other, populations. For example, while the intake of fruit and vegetables is commonly recommended, certain types of these foods may actually promote obesity through the impact the exert on glycemic response which relates to the increase in blood glucose levels after consumption of food [29]. Foods that are digested and absorbed quickly, such as refined grains and potatoes, have a high glycaemic index and should thus be limited, to an extent, in the diet of the growing adolescent [29]. While parents can influence positive consumption of fruits and vegetables in the same way that they can influence breakfast eating, other innovative interventions have been demonstrated to be effective. As an example, McAleese and Rankin [30] reported that a garden-based nutrition intervention, conducted in a school environment, resulted in higher fruit and vegetable consumption in adolescents compared to other two groups in the study. The researchers also reported increases in the intake of vitamin A, vitamin C, and fibre following the intervention which took place in a 25x25 feet garden consisting of two raised strawberry beds, a herb garden and several crops such as potatoes, beans, broccoli, tomatoes, and spinach. During the 12-week study period, the adolescent participants were responsible for maintenance of the garden area and also contributed to the planting and harvesting of the foods they consumed. Though such an intervention may be beyond the available resources of many schools in the United Kingdom, this factor is still readily accessible and implementable with (albeit limited) previous evidence underlining the popularity of growing vegetables in home gardens, even during recessionary times [31, 32].

Though the above recommended strategies can be advised, a key issue relating to their implementation is the level of compliance that is shown by parents and adolescents. It has previously been recommended that interventions should be fun so as to underpin compliance to diet and exercise programmes in children [33] and there are a number of research-supported strategies that can be followed to support this assertion. For example, mobile phone technology can be a useful tool for adolescents with obesity in monitoring any potential weight issues they experience. A study found that using a Fitbit, an online educational programme, and biweekly text messages, significantly improved BMI, blood pressure, physical activity, diet, and self-efficacy among participants [34]. Indeed, such a technology-based intervention could serve as a feasible and effective method for weight management in busy primary care clinics [34]. Gamification and video games have also been shown to be effective tools in enhancing healthy behaviours in children [35]. By incorporating game mechanics, such as points and rewards, into healthy activities, adolescents may be motivated to engage in the behaviours they are designed to promote more frequently. Video games that require physical exercise and movement, such as dance or sports games, can also serve as an enjoyable outlet to increase physical activity [36]. Similarly, nutrition-based games can help in the education of children in relation to healthy food choices and to promote positive attitudes towards healthy eating in general.

Finally, the current findings also suggest that the number of sleep hours on school days, but not on non-school days, may be a lesser influential factor in promoting healthy outcomes in adolescents. This is consistent with previous research showing that adequate sleep is associated with a range of positive health outcomes, such as improved cognitive function, better weight management, and lower risk of chronic diseases. The small, yet significant, effect of sleep on self-reported health serves as a timely reminder to the reader that the synergistic relationship between all of the described healthy behaviours in this study means that a combined strategy to underpinning good health in young populations is warranted. In this way, it should be noted that the small to medium effect sizes reported imply that no one of these behaviours is likely to exert an appreciable effect on self-reported health when carried out in isolation. However, the collective adoption of healthy eating, exercise and sleep habits is likely to be a powerful stimulus to good health in adolescents.

There are limitations to this study. In very large datasets, while a statistical test may demonstrate a significant effect of one variable on another, it is entirely possible that that effect may not hold much practical significance. Statistical significance therefore does not necessarily imply practical significance in the case of the current study and researchers should consider the magnitude of any reported effect sizes, as well as other important factors, such as clinical applicability or economic impact, when translating such findings to practice. In this way, it is crucially important to interpret statistical significance in the context of the research question and the practical significance of the findings of the study [37]. Also, the measure of self-reported health used in this study refers to an individual's subjective evaluation of their own health status, while actual health is determined by objective measures such as clinical assessments, laboratory tests, and medical diagnoses. Accordingly, any measures of perceived health and actual health may not necessarily align, as an individual's perceptions can be impacted by factors such as their mental and emotional state, social support and lifestyle habits. To exemplify this, an individual who feels happy and satisfied with their life may perceive themselves to be healthier than their actual health status indicates despite any objective health tests to the contrary. On the other hand, a person who experiences anxiety or stress may perceive themselves to be less healthy than their actual health status suggests they are. Accordingly, it is vital to consider both perceived and actual health when assessing an individual's overall health status. Lastly, multiple regression analysis may not sufficiently capture the complexities and potentially non-linear interactions of health behaviours in adolescents. Such behaviours are multifaceted and variable, particularly in a dynamically developing population. The use of linear models could oversimplify these behaviours; thus, while our study provides valuable insights, interpretations of the relationships between lifestyle factors and self-reported health are constrained by certain methodological limitations.

## Conclusion

Collectively, the results of this study serve as further support for the importance of adolescents engaging in regular exercise, consuming a healthy breakfast and eating fruits and vegetables, and getting adequate sleep. The findings could hold important implications for public health campaigns that are aimed at promoting healthy behaviours and preventing chronic diseases in this population and beyond that into the longer lifespan. It is important to reemphasise that all of these factors constitute modifiable behaviours that individuals can execute in order to enhance their own personal health; however it is advisable that one engages with all variables at once instead of just one or two as the synergistic effect of combining them could be substantial. An inherent high level of control over these factors suggests that even small improvements in these behaviours could translate into significant health benefits over time when combined.

Accordingly, future research should focus on establishing how long-term compliance can be encouraged, perhaps by incorporating technology and gamification into the execution of healthy life behaviours in children and adolescents thus making such activities more fun and engaging.

## Author Contributions

**Conceptualization:** Jason Moran, Gavin Sandercock, Brandon S. Shaw.

**Data curation:** Jason Moran, Gavin Sandercock.

**Formal analysis:** Jason Moran, Gavin Sandercock.

**Investigation:** Jason Moran, Gavin Sandercock.

**Methodology:** Jason Moran, Gavin Sandercock.

**Project administration:** Jason Moran.

**Resources:** Jason Moran.

**Writing – original draft:** Jason Moran, Gavin Sandercock, Brandon S. Shaw, Paul Freeman, Catherine Kerr, Ina Shaw.

**Writing – review & editing:** Jason Moran, Gavin Sandercock, Brandon S. Shaw, Paul Freeman, Catherine Kerr, Ina Shaw.

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
