## [Decision Letter · Decision Letter 0]

27 Feb 2024

PONE-D-23-36958The Relationship between Easily-Modifiable Lifestyle Habits and Self-Reported Health in Children and Adolescents in the United kingdomPLOS ONE

Dear Dr. Moran,

Thank you for submitting your manuscript to PLOS ONE. After careful consideration, we feel that it has merit but does not fully meet PLOS ONE’s publication criteria as it currently stands. Therefore, we invite you to submit a revised version of the manuscript that addresses the points raised during the review process.

We look forward to receiving your revised manuscript.

Kind regards,

Adedayo Ajidahun

Academic Editor

PLOS ONE

Journal Requirements:

Reviewers' comments:

Reviewer's Responses to Questions

**Comments to the Author**

1. Is the manuscript technically sound, and do the data support the conclusions?

Reviewer #1: Yes

Reviewer #2: Yes

2. Has the statistical analysis been performed appropriately and rigorously? 

Reviewer #1: Yes

Reviewer #2: Yes

3. Have the authors made all data underlying the findings in their manuscript fully available?

Reviewer #1: Yes

Reviewer #2: Yes

4. Is the manuscript presented in an intelligible fashion and written in standard English?

Reviewer #1: Yes

Reviewer #2: Yes

5. Review Comments to the Author

Reviewer #1: Thank you for the opportunity to review this paper who analyzed the relationship between specific easily modifiable lifestyle factors and self-reported health in Children and Adolescents in the United kingdom. The date is from Millennium Cohort Study, specifically the fifth sweep in 2015. The study's utilization of multiple regression analysis provides a robust framework for examining these associations, offering valuable insights into potential avenues for promoting healthier behaviors and preventing chronic diseases.

The authors should move table 1 in the section Results, now is placed before the Statistical Analysis.

Overall, this paper makes a significant contribution to the literature on lifestyle factors and health outcomes in young individuals. However, it would be beneficial for the authors to provide additional insights into the limitations of the study, such as potential confounding variables or biases, which could impact the generalizability of the findings. Additionally, suggestions for future research directions would further enhance the paper's contribution to the field.

The references should be check by the authors as some of them are not in the Journal format and they are written in different styles.

Reviewer #2: The manuscript provides the findings of a study conducted with the objective to determine the interdependence of modifiable lifestyle factors on self-reported health in British adolescents. The theoretical basis is well founded and justifies the study. The bibliographic references are current and relevant to the subject of study. The methods used are sufficiently described, appropriate and consistent with the study proposal. The proposal is presented in an attractive manner, and the discussion of the findings show good prospects to the knowledge area. The conclusions are consistent with presented evidence and arguments. The findings of the study have good prospects to offer important contributions to the knowledge area, and offers important support for the design and conduct of future studies.

However, with the intention of helping to improve the manuscript, I suggest adjusting two items:

(a) Table 1 - Descriptive Statistics for Participant Characteristics: data related to self-reported health and modifiable lifestyle behaviors should not be presented through mean and standard deviation, but rather through prevalence rates (%) and respective confidence intervals (CI 95%) for each categorical answers.

(b) The use of the term "Easily-Modifiable Lifestyle Habits": I understand and studies have shown that it is eventually possible to modify lifestyle habits, particularly in childhood and adolescence. However, this is an "extremely difficult" task, and not as the authors mention, "easily". Furthermore, there are important conceptual differences between "Habits" and "Conducts". Therefore, specifically in this study, I believe that the use of the term "Easily-Modifiable Lifestyle Habits" at various times in the text is not the most appropriate. I suggest replacing the term with "Modifiable Lifestyle Behaviors".

6. PLOS authors have the option to publish the peer review history of their article (what does this mean?). If published, this will include your full peer review and any attached files.

Reviewer #1: No

Reviewer #2: No

---

## [Author Response · Author response to Decision Letter 0]

11 Apr 2024

Reviewer #1: Thank you for the opportunity to review this paper who analyzed the relationship between specific easily modifiable lifestyle factors and self-reported health in Children and Adolescents in the United Kingdom. The date is from Millennium Cohort Study, specifically the fifth sweep in 2015. The study's utilization of multiple regression analysis provides a robust framework for examining these associations, offering valuable insights into potential avenues for promoting healthier behaviors and preventing chronic diseases.

Response: We would like to sincerely thank the reviewer for the time they have taken to review our manuscript. We greatly appreciate the recommendations that have been made and we have responded in an itemised fashion in this document, highlighting the changes in the manuscript.

Reviewer #1:The authors should move table 1 in the section Results, now is placed before the Statistical Analysis.

Response: We have now added this to results section.

Reviewer #1: Overall, this paper makes a significant contribution to the literature on lifestyle factors and health outcomes in young individuals. However, it would be beneficial for the authors to provide additional insights into the limitations of the study, such as potential confounding variables or biases, which could impact the generalizability of the findings.

Response: We do have an already extensive limitations section but have now added:

“Lastly, multiple regression analysis may not sufficiently capture the complexities and potentially non-linear interactions of health behaviours in adolescents. Such behaviours are multifaceted and variable, particularly in a dynamically developing population. The use of linear models could oversimplify these behaviours; thus, while our study provides valuable insights, interpretations of the relationships between lifestyle factors and self-reported health are constrained by certain methodological limitations.”

Reviewer #1: Additionally, suggestions for future research directions would further enhance the paper's contribution to the field.

Response: We have now added:

“Accordingly, future research should focus on establishing how long-term compliance can be encouraged, perhaps by incorporating technology and gamification into the execution of healthy life behaviours in children and adolescents thus making such activities more fun and engaging.”

Reviewer #1: The references should be check by the authors as some of them are not in the Journal format and they are written in different styles.

Response: We have now standardised and corrected the references.

Reviewer #2: The manuscript provides the findings of a study conducted with the objective to determine the interdependence of modifiable lifestyle factors on self-reported health in British adolescents. The theoretical basis is well founded and justifies the study. The bibliographic references are current and relevant to the subject of study. The methods used are sufficiently described, appropriate and consistent with the study proposal. The proposal is presented in an attractive manner, and the discussion of the findings show good prospects to the knowledge area. The conclusions are consistent with presented evidence and arguments. The findings of the study have good prospects to offer important contributions to the knowledge area, and offers important support for the design and conduct of future studies.

Response: We are very grateful to the reviewer for dedicating the time to critiquing our manuscript. The insightful suggestions have been valuable and we have carefully addressed each one, noting the corresponding modifications within the manuscript.

Reviewer #2: However, with the intention of helping to improve the manuscript, I suggest adjusting two items:

Table 1 - Descriptive Statistics for Participant Characteristics: data related to self-reported health and modifiable lifestyle behaviors should not be presented through mean and standard deviation, but rather through prevalence rates (%) and respective confidence intervals (CI 95%) for each categorical answers.

Response: We have now replaced Table 1 with the frequency table from our statistical analysis, altered for publication standards.

Reviewer #2: The use of the term "Easily-Modifiable Lifestyle Habits": I understand and studies have shown that it is eventually possible to modify lifestyle habits, particularly in childhood and adolescence. However, this is an "extremely difficult" task, and not as the authors mention, "easily". Furthermore, there are important conceptual differences between "Habits" and "Conducts". Therefore, specifically in this study, I believe that the use of the term "Easily-Modifiable Lifestyle Habits" at various times in the text is not the most appropriate. I suggest replacing the term with "Modifiable Lifestyle Behaviors".

Response:

We agree fully with the reviewer and we have adjusted the language used throughout the text. We now use: ‘modifiable lifestyle behaviours’ as per the reviewer’s recommendation.

---

## [Decision Letter · Decision Letter 1]

29 Apr 2024

The Relationship between Modifiable Lifestyle Behaviours and Self-Reported Health in Children and Adolescents in the United Kingdom

PONE-D-23-36958R1

Dear Dr. Moran,

We’re pleased to inform you that your manuscript has been judged scientifically suitable for publication and will be formally accepted for publication once it meets all outstanding technical requirements.

Kind regards,

Adedayo Ajidahun

Academic Editor

PLOS ONE

Additional Editor Comments (optional):

Reviewers' comments:

Reviewer's Responses to Questions

**Comments to the Author**

1. If the authors have adequately addressed your comments raised in a previous round of review and you feel that this manuscript is now acceptable for publication, you may indicate that here to bypass the “Comments to the Author” section, enter your conflict of interest statement in the “Confidential to Editor” section, and submit your "Accept" recommendation.

Reviewer #1: All comments have been addressed

Reviewer #2: All comments have been addressed

2. Is the manuscript technically sound, and do the data support the conclusions?

Reviewer #1: Yes

Reviewer #2: Yes

3. Has the statistical analysis been performed appropriately and rigorously? 

Reviewer #1: Yes

Reviewer #2: Yes

4. Have the authors made all data underlying the findings in their manuscript fully available?

Reviewer #1: Yes

Reviewer #2: Yes

5. Is the manuscript presented in an intelligible fashion and written in standard English?

Reviewer #1: Yes

Reviewer #2: Yes

6. Review Comments to the Author

Reviewer #1: The authors had reviewed the paper and answer all the comments from the reviewers and in mu opinion the paper is ready to be publish.

Reviewer #2: I thank the authors for considering the aspects highlighted in the first review and for the effort to attend to them. All punctuated questions were answered satisfactorily by the authors, leaving no further observation to be made in the last version presented.

7. PLOS authors have the option to publish the peer review history of their article (what does this mean?). If published, this will include your full peer review and any attached files.

Reviewer #1: No

Reviewer #2: **Yes: **Dartagnan Pinto Guedes (State University of Northern Parana, Brazil)

---

## [Editor Report · Acceptance letter]

2 May 2024

PONE-D-23-36958R1 

PLOS ONE

Dear Dr. Moran, 

I'm pleased to inform you that your manuscript has been deemed suitable for publication in PLOS ONE. Congratulations! Your manuscript is now being handed over to our production team.

Kind regards, 

on behalf of

Dr. Adedayo Ajidahun 

Academic Editor

PLOS ONE